# Disentangling Physical Forcings Influencing Exchange Flow in a Multi-basin Fjord System: Chiloé Inner Sea, Patagonia

Elias Pinilla<sup>1,2</sup> and Lauren M. Ross<sup>1</sup>

**Correspondence:** Elias Pinilla (elias.pinilla@maine.edu)

Abstract. Fjords mediate land-sea exchange of water masses, nutrients, and dissolved oxygen in high latitude coastal areas. The renewal of fjord waters is sensitive to shifts in runoff, winds, and tides that could be exacerbated due to climate change. In this study we quantify the relative contribution of rivers, wind, tides, and boundary temperature and salinity to estuarine exchange flow in the multi-basin Chiloé Inner Sea (CIS). We use a suite of controlled 3D simulations that toggle each driver on/off and analyze the outputs with the Total Exchange Flow (TEF) framework. TEF separates exchange inflow and outflow via salinity classes, allowing salt import to be a proxy for renewal potential. Idealized experiments isolating exchange flow drivers show that internal baroclinicity from rivers sets the baseline two-layer exchange flow and explains  $\sim 45-55\%$  of the variability of the exchange inflow  $(Q_{in})$ , the dynamic stratification  $(\Delta S)$ , and the salt import  $(Q_{in}\Delta S)$  along the CIS. Wind modulates the baseline exchange flow by ( $\sim 20-50\%$ ), enhancing  $Q_{\rm in}$  when wind stress is down–estuary while strong wind of either sign erodes  $\Delta S$  and thus salt import. Tides contribute less on average than wind to the exchange inflow ( $\sim$ 10-20%) but their influence changes spatially. In interior basins the influence of tides is minimal, while at sills and channel constrictions spring tides tend to increase  $Q_{\rm in}$  and often  $Q_{\rm in}\Delta S$  without a proportional collapse of  $\Delta S$ . Offshore temperature and salinity variability, also called boundary baroclinicity, is of secondary influence, yet is consequential near the CIS mouth in winter, and can at times weaken or reverse the internally driven exchange. Renewal of deep fjord waters requires both ample exchange inflow and dynamic stratification. Our results indicate that in the CIS, the likelihood of salt import is greatest under moderate down-estuary winds coincident with high runoff and spring tides, whereas persistent up-estuary winds or strong wind events suppress import despite large  $Q_{\mathrm{in}}$ . The exchange inflow along the CIS was found to have dominant synoptic (3–12 d; wind/storms), monthly (25–35) and intraseasonal (40–70 d) periodicities that are driven primarily by atmospheric phenomena in the Southern Hemisphere. These results highlight new mechanisms and periodicities important for deep-water renewal events to take place, and therefore can help anticipate when renewal will be suppressed and low dissolved oxygen could become a risk in the fjords and channels of the CIS.

#### 1 Introduction

Fjords are long, narrow, and deep glacially-carved valleys flooded by the sea during the last deglaciation and commonly segmented by submarine sills (Inall and Gillibrand, 2010). Their geometry and substantial freshwater inputs promote persistent

<sup>&</sup>lt;sup>1</sup>Department of Civil and Environmental Engineering, University of Maine, Orono, USA.

<sup>&</sup>lt;sup>2</sup>Environmental Department, Fisheries Development Institute (IFOP), Chiloé, Chile.

stratification and a tidally averaged, two-layer exchange flow in which fresher surface waters exit while saltier waters enter at depth. This exchange is known to regulate ventilation, water renewal, and biogeochemical pathways in semi-enclosed basins (Allen et al., 2025; MacCready et al., 2021; Geyer and MacCready, 2014). Beyond river discharge, a growing body of observational, analytical, and numerical work shows that tides, winds, and remote oceanic variability can act as a first-order control on exchange flow, with relative importance that shifts across space and time (Lange et al., 2020; Sanchez et al., 2025; Jackson et al., 2018; Arneborg et al., 2004; Soto-Riquelme et al., 2023; Giddings and MacCready, 2017). Tides influence exchange through wave-driven residuals, sea-level set-up/set-down, spatial gradients of tidal currents, and, indirectly, through mixing that reshapes vertical and along-channel density structure (Valle-Levinson, 2022; MacCready and Geyer, 2024). Winds act locally through isohaline straining, whereby along-channel winds interact with background density gradients to modify stratification, enhancing stratification with down-estuary winds and reducing it with up-estuary winds by enhancing turbulence (stirring) (Scully et al., 2005). Winds can also drive depth-uniform residuals via barotropic set-up (Csanady, 1973; Chen and Sanford, 2009; Valle-Levinson and Blanco, 2004). Shelf winds and density anomalies have also been found to further transmit low-frequency pressure signals into fjords (Ross et al., 2025; Pérez-Santos et al., 2019; Giddings and MacCready, 2017; Jackson et al., 2018; Inall et al., 2015; Arneborg et al., 2004).

Exchange flow in fjords arises from the combined action of winds, tides, river discharge, and remote ocean variability, with each forcing imprinting characteristic time scales on the circulation. Over synoptic weather bands ( $\sim$ 3–12 d), atmospheric lowpressure systems enhance wind stresses that frequently align with pulsed river discharges that overlay their broader seasonal envelopes (Soto-Riquelme et al., 2023; Giddings and MacCready, 2017; Valle-Levinson and Blanco, 2004). On fortnightly bands (~14-15 d), the spring-neap cycle imposes a modulation of tidal amplitude and currents (Geyer and MacCready, 2014; MacCready and Geyer, 2024). Over monthly bands ( $\sim$ 25–35 d), in the Southern Hemisphere mid-latitude storm–track variability, often summarized by the Baroclinic Annular Mode (BAM) (Thompson and Woodworth, 2014; Ross et al., 2015), manifests across diverse atmospheric and oceanic variables and phenomena in Patagonian fjords (e.g., currents, temperature, and salinity, wind stresses) (Ross et al., 2015; Pérez-Santos et al., 2019; Narváez et al., 2019). At longer intraseasonal bands (~40-70 d), the Madden-Julian Oscillation has been found to drive coastally trapped waves around continental margins and influence atmospheric forcing such as wind stress (Poli et al., 2022) that can influence the offshore forcing contribution to fjords (Ross et al., 2025). River discharge superposes seasonal cycles with storm-driven pulses at synoptic to weekly scales in many fjord catchments, adding low-frequency variability to the forcing set (Inall and Gillibrand, 2010; Sutherland et al., 2011). Due to the complexity and large size of fjord and channel systems it is often difficult to diagnose drivers of circulation patterns as well as their frequency of variability. Few studies have attempted this, and have found the importance of tides, winds and rivers in driving circulation over all of the above-referenced periodicities, but these studies have focused on single points and limited time periods (Ross et al., 2025; Pinilla et al., 2024).

The Chiloé Inner Sea (CIS) of northern Patagonia (41–44°S) is a multi-basin fjord and channel system connected to the Pacific Ocean by a broad mouth and segmented by sills and narrows. It experiences strong seasonality in freshwater input, energetic tides, vigorous winds, and time-varying ocean boundary conditions (Narváez et al., 2019). Previous studies have documented the impact of wind on stratification and exchange (Soto-Riquelme et al., 2023; Pinilla et al., 2024; Valle-Levinson

65

75

and Blanco, 2004; Castillo et al., 2016), sill–controlled tidal dynamics including multi–layer residuals (Cáceres et al., 2003; Valle-Levinson et al., 2014, 2007; Castillo et al., 2012; Pinilla et al., 2020a), and pronounced seasonal shifts in density structure (Ruiz et al., 2021). Yet a system–wide decomposition of exchange flow drivers across its multi-basins remains unknown. These attributes make the CIS an ideal natural laboratory for disentangling how tides, winds, rivers, and remote baroclinicity shape exchange flows across space and seasons.

In this study we combine a realistic 3-dimensional model hindcast of the CIS with targeted switch—off/on numerical experiments to quantify exchange flows using the Total Exchange Flow (TEF) framework. The TEF methodology partitions transport by salinity space and yields physically interpretable measures consistent with the classical salt budget (MacCready, 2011; Burchard et al., 2018, 2019). By quantifying the TEF inflow and the dynamic stratification, defined as the salinity difference between the outflowing and inflowing layers, their product yields the net salt flux entering the fjord. This net salt input displaces resident, fresher deep water and thus constitutes the primary advective mechanism driving water renewal and ventilation (Allen and Simpson, 1998; Stigebrandt, 2012; Johnsen et al., 2024; Sagen et al., 2025). Using experiments that isolate four drivers: salinity and temperature at the ocean boundary (external baroclinicity), rivers (internal baroclinicity), tides, and wind, we ask: (1) How do driver contributions to TEF vary along the CIS?, (11) How do they change temporally, including spring—neap cycles and seasonally?, and (111) Under what conditions does salt flux increase and decrease in relation to TEF and its drivers along the CIS?

The remainder of this manuscript is organized as follows: Section 2 describes the model setup, the atmospheric, oceanic, river forcings, and the cross-sectional TEF metrics. It also defines the four idealized numerical experiments, referred to as 'switch' experiments as forcing mechanisms are switched 'on' and 'off'. Section 3 begins by outlining the typical hydro-climatic setting of the CIS, then presents year-long subtidal time series of TEF metrics at three considered cross sections along the CIS. This is followed by a spatial comparison of the impact of the isolated forcing mechanisms on the TEF parameters. Section 4 discusses how, and over what dominant periodicities, tides, winds, and baroclinicity can vary exchange flows and links these patterns to dominant modes of climate variability. Finally, Section 5 states the conclusions.

## 2 Methods

# 2.1 Study area

The Chiloé Inner Sea (CIS; 41–44° S; Fig. 1) is a stratified, multi-basin fjord complex in northern Patagonia that extends ~260 km from Reloncaví Sound in the north to Corcovado Gulf in the south. It connects to the Pacific Ocean through two passages: the broad (~40 km), deep (~200 m) Boca del Guafo to the south, the main gateway for oceanic exchange, and the narrower (~3 km, ~100 m) and tidally energetic Chacao Channel to the north (Cáceres et al., 2003). Interior basins (Reloncavi Sound, Ancud and Corcovado Gulfs, ~50 km wide) are segmented by sills and narrows, notably the Desertores Pass, and are flanked by steep fjords (e.g., Reloncaví, Comau). Typical depths are ~200 m in the main channels, locally exceeding ~450 m in fjord basins (Narváez et al., 2019).

**Figure 1.** Study area in the context of the (a) world and (b) Patagonia, southern end of South America. (c) Hydrodynamic model domain with the unstructured mesh overlain, centered on Chiloé Inner Sea. Colors show bathymetry (m). Dashed segments mark cross—sections used for TEF diagnostics that are referred to as the Corcovado (southernmost), Desertores (middle) and Reloncavi (uppermost) transects. The major sub—basins and rivers that contribute freshwater to the study area are labeled.

Key forcings vary strongly in space and time. The tides are predominantly semidiurnal (dominated by M2) and amplify landward by barotropic resonance: ranges grow from  $\sim$ 1.5–2 m at Boca del Guafo to 6–7 m in the Reloncav Sound, with currents up to  $\sim$  4 m, s<sup>-1</sup> in constrictions such as Chacao (Aiken, 2008; Cáceres et al., 2003). Tidal current amplitude at the sills is modulated by the spring–neap cycle ( $\approx$ 14.8 d) and local geometry (Artal et al., 2019). Freshwater inputs from Andean rivers (e.g., Puelo, Yelcho; annual mean combined  $\sim$ 1300 m<sup>3</sup> s<sup>-1</sup>) and high precipitation sustain a low-salinity surface layer

100

105

115

120

with winter–spring peaks (Aguayo et al., 2019). Winds are seasonally organized (southerly/southwesterly in spring–summer; northerly in fall–winter), punctuated by 2–12 d synoptic events that modulate shear, outflow, and occasionally residuals (Soto-Riquelme et al., 2023; Pinilla et al., 2024, 2020b). At the ocean boundary, shelf variability enters via Boca del Guafo: barotropic pressure and density anomalies at  $\sim$ 23–30 d and  $\sim$ 40–70 d have been found to alter inflow properties and stratification (Ross et al., 2025; Mundaca et al., 2025; Strub et al., 2019).

#### 2.2 Model setup

We simulate the CIS circulation with a hydrostatic, baroclinic MIKE 3 FM model on an unstructured mesh that resolves the Boca del Guafo, the interior basins and sills of the CIS, up to the heads of major fjords (Fig.1). Horizontal grid resolution varies from  $\sim$ 200 to 1000 m. The vertical grid has 53 hybrid  $\sigma$ –z layers with <2 m near-surface spacing to capture stratification. Bathymetry merges Hydrographic and Oceanographic Service of the Chilean Navy (SHOA) charts and local surveys. Turbulence closure includes the standard k– $\varepsilon$  scheme (Launder and Spalding, 1974; Rodi, 1984).

Surface momentum, pressure, and heat fluxes come from a WRF-IFOP hindcast at 3 km resolution, which has been evaluated in the region (wind/pressure correlations > 0.8; Soto-Riquelme et al., 2023). River inflows are prescribed from the FLOW-IFOP hydrological model (e.g., Puelo, Vodudahue, Yelcho) (Reche et al., 2021). Astronomical tides enter as harmonic elevations at the open boundary (Pawlowicz et al., 2002) using a validated regional barotropic solution (Pinilla et al., 2012). At the open boundary, we prescribe a bimonthly temperature-salinity climatology from CIMAR-Fiordos CTD profiles (Carrasco and Silva, 2010; Valdenegro, 2010). It reproduces the dominant seasonal baroclinic cycle, slightly fresher waters with a deeper eroded pycnocline in winter and a sharper pycnocline in summer (Ross et al., 2025; Ruiz et al., 2021). This seasonal boundary omits synoptic and mesoscale variability; we note this as a limitation, but use it to better isolate the interior effects of tides, wind, and rivers. Bimonthly fields are linearly interpolated in time. After a two-year spin-up, we model the years from 2015–2018, yet focus on all analyzes for this manuscript from 2018. We focus on 2018 because it spans a wide range of tidal, wind, river, and boundary conditions, enabling robust driver-TEF relationships without tying the analysis to specific events. Moreover, 2018 is representative of prevailing wind magnitudes/directions and river discharges in the CIS (Pinilla et al., 2024). The model used in this study is an iteratively improved version used in Linford et al. (2023); Landaeta et al. (2023); Mardones et al. (2021); Pérez-Santos et al. (2021). The model has been extensively validated against salinity, current velocities, wind velocities, river discharge and water level with more information on the validation presented in the Supplementary Information accompanying this manuscript.

# 2.3 Total Exchange Flow (TEF)

The TEF framework quantifies exchange by sorting transport in *salinity space* rather than fixed-depth layers, a choice motivated by the unsteady, continuously varying salinity structure of realistic estuaries (Maccready et al., 2018); this yields a compact, two-branch equivalent that is robust to variability (MacCready, 2011; Burchard et al., 2018). At each section (Corcovado, Desertores, Reloncaví; Fig. 1) we compute cumulative transport versus salinity and locate the *dividing salinity* where it changes sign, which separates the landward (denser) from the seaward (fresher) branch (extended dividing-salinity method; Lorenz

et al., 2019). This construction provides physically interpretable quantities used throughout (see MacCready 2011; Lorenz et al. 2019; Burchard et al. 2018 for details):

## 2.3.1 Exchange Inflow and outflow:

 $Q_{\rm in}$  is the volumetric transport of the landward (denser) branch;  $Q_{\rm out}$  is the volume transport of the seaward (fresher) branch.

## 2.3.2 Layer-mean salinities:

 $S_{
m in}$  and  $S_{
m out}$  are transport-weighted salinities of the inflow  $(Q_{
m in})$  and outflow  $(Q_{
m out})$  branches. Their difference:

 $\Delta S = S_{\rm in} - S_{\rm out}$ , measures how distinct the two branches are. We refer to  $\Delta S$  as the dynamic stratification because it reflects the salinity contrast between the actively exchanging layers.

## 2.3.3 Landward salt flux by exchange:

The net salt carried landward by the exchange inflow is:  $Q_{\rm in} \Delta S$ . This is the product of how much water enters ( $Q_{\rm in}$ ) and how salty the inflow ( $\Delta S$ ) is relative to the outflow (Burchard et al., 2018; Broatch and Maccready, 2022). Increases in tidal or wind energy may raise  $Q_{\rm in}$  yet simultaneously reduce  $\Delta S$  through mixing, so  $Q_{\rm in} \Delta S$  can either grow or decline depending on which effect dominates (e.g., MacCready and Geyer, 2024).

We compute  $Q_{\rm in}$ ,  $Q_{\rm out}$ ,  $S_{\rm in}$ ,  $S_{\rm out}$  and  $\Delta S$  hourly on each model section by binning transports in salinity space using 1000 evenly spaced bins over the full salinity range of the model output following MacCready (2011) and Lorenz et al. (2019).

#### 145 2.4 Numerical experiments

We isolate forcings by running the same model with different parameters switched 'on' and 'off', as is shown in Table 1. The different combinations of experiments include a first with baroclinicity, tides, and wind that will be referred to as **BTW**. The baroclinicity term includes both the contribution of rivers, which we refer to as the internal (I) influence of baroclinity, and external (E), which is from the models ocean boundary forced with a climatology of temperature and salinity (T-S) from CTD casts. The second combination is internal and external baroclinicity B(I+E), tides but with no wind forcing (**BT**). The third combination is only baroclinic (internal and external forcing) and no wind or tides (**B(I+E)**). Finally, we isolate only the external baroclinic forcing to determine the importance of changes in temperature and salinity in the ocean on the exchange flow in the fjords (**B(E)**).

## 155 2.4.1 Driver decomposition

For any TEF metric  $Y \in \{Q_{\rm in}, \Delta S, Q_{\rm in} \Delta S\}$ , we isolate each driver's total effect (direct + interactions) by differencing model experiments that are identical except for that driver:

**Table 1.** Description of model experiments with different forcing mechanisms turned on and off. The term 'B' refers to Baroclinic, with 'I' indicating internal baroclinic contributions (river) and 'E' external contributions from the open ocean boundary, 'T' is tides, and 'W' is wind.

| Experiment                      | Tides        | Wind | Rivers       | Boundary $T$ – $S$ |
|---------------------------------|--------------|------|--------------|--------------------|
| BTW (Full)                      | <b>√</b>     | ✓    | ✓            | ✓                  |
| BT (No wind)                    | $\checkmark$ | _    | $\checkmark$ | $\checkmark$       |
| B(I+E) (Baroclinic)             | _            | _    | $\checkmark$ | $\checkmark$       |
| B(E) (External Baroclinic only) | _            | _    | _            | $\checkmark$       |

$$Y^{\text{wind}} = Y_{\text{BTW}} - Y_{\text{BT}},\tag{1}$$

$$Y^{\text{tide}} = Y_{\text{BT}} - Y_{\text{B}},\tag{2}$$

$$160 Y^B = Y_B, (3)$$

$$Y^{B(E)} = Y_{B(E)}. (4)$$

Here, B denotes the baroclinic run with internal+external buoyancy B(I+E).

Our switch experiments (Table 1) estimate the total effect of each forcing, its direct influence plus co-acting interactions (non-linear interaction), rather than additive components that sum up to the full signal. This design yields practical, comparable attribution of each forcings contribution (inclusive of interactions) to the TEF metrics. For example, Guo and Valle-Levinson (2008) shows that wind–tide–buoyancy coupling violates simple additivity and further demonstrate that subtracting the density-driven (no-wind) exchange from wind modified fields is not equivalent to applying wind to a constant-density estuary.

To compare how each forcing modulates exchange, we define specific indices per driver, denoted X(Table 2):

- 170 (i) **Tides**: A dimensionless spring—neap index  $(I_{SN})$  scaled to [0,1] (0 = neap, 1 = spring).
  - (ii) **Wind**: The north–south surface stress,  $\tau_y$  (N m<sup>-2</sup>), obtained from WRF winds for each section (Reloncaví, Desertores, Corcovado); by convention,  $\tau_y > 0$  is northward (up–estuary).
  - (iii) **Baroclinicity**: the along-estuary salinity gradient  $(\partial_x S)^B$  from Reloncaví to Corcovado cross-sections, computed from the baroclinic run B(I+E) using the model depth-mean salinity,  $(\partial_x S)^B = \left[\overline{S}^B(\text{seaward}) \overline{S}^B(\text{landward})\right]/\Delta x$ . Values are positive when salinity increases seaward.

A simple analysis tool to show how a TEF metric Y behaves as a forcing X increases is to group X into ranges and, for each range, summarize the concurrent Y values by their median and spread. We will refer to this diagnostic as a *driver-response curve*. Concretely, given paired observations (X,Y) sampled N times (e.g., hourly data). We split the observed values of X into X bins with similar sample counts. For each bin  $B_k$ , we compute the median of Y (typical value), the interquartile range (IQR, 25–75%) of Y (spread), and we plot the median of Y against the median of Y with IQR bars. This answers, directly from the data, when the driver is low/medium/high, what is a typical Y and how variable is it? while remaining robust to outliers.

**Table 2.** Driver indices X used in the driver–response analysis.

| Driver        | Index                  | Definition / computation                                                                                                                                                                                          | Units                             | Sign / convention                         |
|---------------|------------------------|-------------------------------------------------------------------------------------------------------------------------------------------------------------------------------------------------------------------|-----------------------------------|-------------------------------------------|
| Tides         | $I_{\rm SN} \in [0,1]$ | Spring–neap index evaluated at each section; $I_{\rm SN}=0$ at neap and $I_{\rm SN}=1$ at spring.                                                                                                                 | -                                 | 7                                         |
| Wind          | $	au_y$                | Along-estuary surface stress at each section.                                                                                                                                                                     | ${ m Nm^{-2}}$                    | $\tau_y > 0$ : northward (up-estuary).    |
| Baroclinicity | $(\partial_x S)^B$     | Horizontal salinity gradient computed from $B(I+E)$ : $(\partial_x S)^B = \overline{S}^B (\text{seaward}) - \overline{S}^B (\text{landward})]/\Delta x$ . Here $\overline{S}^B$ is the model depth-mean salinity. | $10^{-3}\mathrm{gkg^{-1}km^{-1}}$ | Positive when salinity increases seaward. |

*Notes:* B denotes the baroclinic run with internal+external buoyancy B(I+E).

## 3 Results




## 3.1 Exchange-flow forcing in the Chiloé Inner Sea: broad-scale context

We first outline the typical conditions of the hydro-climatic drivers in the CIS: freshwater input, wind stress and tidal-current amplitude, using 2018 as a representative year. Total river discharge peaks in austral winter–spring (June–November) at  $2.0-2.5 \times 10^3 \,\mathrm{m}^3 \,\mathrm{s}^{-1}$ , fed mainly by the Puelo, Petrohué, Vodudahue and Yelcho rivers (Reloncaví and Corcovado watersheds; Fig. 1), and falls below  $0.5 \times 10^3 \,\mathrm{m}^3 \,\mathrm{s}^{-1}$  in summer (January–February) (Fig. 2a). The annual and depth-averaged density is  $1025-1026 \,\mathrm{kg} \,\mathrm{m}^{-3}$  inside the basins, with lighter water in northern Reloncaví Sound than in the southern Corcovado Gulf (Fig. 2b). The annual and depth-averaged salinity mirrors the density structure, decreasing landward from  $\sim 33.5 \,\mathrm{g/kg}$  near Boca del Guafo to  $







the Austral winter (May–August) (Fig. 3a,b). Overall, in the Austral summer (January–March) the mean wind stress is less than in the Austral winter. Further, the wind-stress forcing exhibits a seasonal reversal with northward (up-estuary) stresses  $< 0.1 \,\mathrm{N\,m^{-2}}$  dominating in summer (January–March), while southward stresses prevail in fall–winter (May–August).

Semidiurnal ( $M_2$ , period of 12.42h) currents exceed  $0.8 \,\mathrm{m\,s^{-1}}$  in the main constrictions of the domain, with peak values in Chacao Channel surpassing  $1 \,\mathrm{m\,s^{-1}}$ , Fig. 3c. The semidiurnal tidal current amplitudes in Desertores Pass and the Corcovado Gulf reach  $0.8-1.0 \,\mathrm{m\,s^{-1}}$ , while currents stay  $< 0.1 \,\mathrm{m\,s^{-1}}$  over the deeper interior basins and fjords (Ancud Gulf, Reloncaví Sound, Comau and Reloncaví fjords). Consequently, tidal energy input is strongest at the two major narrows (Chacao Channel and Desertores Pass) and across the extensive, topographically-steepened margins of Corcovado Gulf, whereas the inner basins experience markedly lower energy levels.

Together these patterns set the stage for exchange flow with a seasonal baroclinicity that strengthens in spring and relaxes in winter, and a sharp spatial contrast in tidal hydraulics which are strong at the mouth and narrows, weak in the interior. We now quantify the TEF using various combinations of the above-described forcing mechanisms (Table 1), and begin by describing the subtidal time series of TEF inflow  $(Q_{in})$ ,  $\Delta S$ , and salt flux  $Q_{in}\Delta S$  across experiments by section.

## 210 3.2 Subtidal variability and annual forcing budget

The subtidal evolution of the exchange flow for the four numerical experiments, as shown in Figure 4, reveal that at the mouth section (Corcovado; Figure 4g) the full-forcing run (BTW, black) exhibits the largest magnitude and fluctuations, with  $Q_{\rm in}$ repeatedly exceeding  $1 \times 10^5$  m<sup>3</sup> s<sup>-1</sup> during spring (October–December) and late fall (April–June) and transiently dropping below the tide plus river curve (BT, orange) under episodic wind reversals (e.g., December and January). The baroclinic run B(I+E) (blue), considering internal and external density forcing, supplies a smooth seasonal baseline that closely tracks the BTW mean at each section considered (Figure 4a,d,g). The external baroclinicity simulation B(E) (green) remains generally near zero with occasional pulses, evidencing that while the bulk of the exchange is generated inside the domain, external signals can propagate inland with notable impacts in winter (July-August) as seen in Corcovado (Figure 4g) and Desertores (Figure 4d). Desertores (middle row of Figure 4) behaves as a damped version of Corcovado with the same bi-modal seasonal pattern present, but peak  $Q_{\rm in}$  values are roughly halved and wind are less pronounced than in Corcovado. In the landward basin (Reloncaví; top row) volume fluxes are one order of magnitude smaller than Corcovado ( $Q_{\rm in}=1$ – $2\times10^4$  m $^3$  s $^{-1}$ ), yet the salinity contrast  $\Delta S$  reaches its maximum (up to  $4\,\mathrm{g\,kg^{-1}}$  in Oct–November, right at the time when river discharges are at their maximum. Figure 4a,b). Consequently, the salt flux into the basin,  $Q_{\rm in}\Delta S$ , peaks at Corcovado (>  $1\times10^5$  g kg $^{-1}$  m $^3$  s $^{-1}$ ) primarily because of the larger  $Q_{\mathrm{in}}$  there; however, when expressed relative to local section, Reloncaví exhibits strong seasonal compensation (winter wind increases  $Q_{\rm in}$  while spring stratification raises  $\Delta S$ ), yielding comparatively high salt import over the year. Note that axis ranges differ by section to preserve subtidal structure without obscuring landward variability.

For each section we decomposed the subtidal TEF series by model contribution (E, I, T, W) and expressed the RMS magnitude of each driver as a percentage of the total (Fig. 5). For the exchange inflow  $Q_{\rm in}$  (panel a), the internal baroclinic (river) contribution dominates at all sections ( $\sim$ 45–55%). Wind provides a secondary share ( $\sim$ 15–25%), slightly weaker at Desertores than at Reloncaví or Corcovado. Tides account for  $\sim$ 10–20% with only a weak changes along the sections, while the external

Figure 2. Hydrographic and Hydrodynamic conditions in the Chiloé Inner Sea during 2018, derived from model simulations. (a) River discharge time series, for groups of rivers contributing to different sections: Reloncaví section (blue, including the Puelo River as the major contributor), Desertores section (red, rivers discharging between the Reloncaví and Desertores sections), Corcovado section (green, rivers discharging between the Desertores and Corcovado sections), with the total summed discharge ( $Q_R$ ) Total shown in black. Thick lines represent 10-day low-pass filtered values, while thin lines indicate daily discharge. (b) Spatial map of annual (2018) mean depth-averaged density (kg m<sup>-3</sup>). (c) Spatial map of annual mean depth-averaged salinity. (d) Spatial map of annual mean stratification index, calculated as the difference between depth-averaged salinity in the surface layer (0–25 m) and the bottom layer (25 m to bottom). (e) Time series of the horizontal salinity gradient between Reloncavi a Corcovado section, derived from depth-averaged salinity, with positive values indicating increasing salinity seaward.


Figure 3. (a) Mean wind stress (N  $m^{-2}$ ) for January–March 2018 (Austral summer), with vectors indicating direction and magnitude. (b) Mean wind stress for May–August 2018 (Austral winter). (c) M2 tidal current amplitude (m  $s^{-1}$ ).

baroclinic signal increases southward (from a few percent at Reloncaví to  $\sim$ 20–25% at Corcovado). For  $\Delta S$  (panel b), rivers remain dominant at all locations ( $\sim$ 50%), with wind second: it is most influential in the inner basin (Reloncaví,  $\approx$ 30–35%) and weakens toward the south ( $\sim$ 20–25%). The tidal contribution to  $\Delta S$  is modest ( $\sim$ 10–15%), and the external baroclinic influence is the smallest (

Figure 4. Subtidal TEF time series by section in 2018. Rows: Reloncaví (a–c), Desertores (d–f), Corcovado (g–i). Columns: inflow magnitude  $Q_{\rm in}$  (left), layer contrast  $\Delta S$  (center), and salt import  $Q_{\rm in}\Delta S$  (right). Colors denote experiments BTW (tides+wind+rivers), BT (tides+rivers), B(I+E) (Internal rivers with External boundary T-S), and B(E) (External boundary T-S only).

share is small at Reloncaví (




Figure 5. Annual mean contributions of each driver—E (external baroclinicity, purple), I (internal baroclinicity/rivers, blue), T (tides, green), W (wind, orange)—as percentages of the total magnitude for (a)  $Q_{\rm in}$  (m<sup>3</sup> s<sup>-1</sup>), (b)  $\Delta S$  (g kg<sup>-1</sup>), and (c)  $Q_{\rm in}\Delta S$  (g kg<sup>-1</sup> m<sup>3</sup> s<sup>-1</sup>) at Reloncaví, Desertores, and Corcovado sections.

## 240 3.3 Effects of baroclinicity on exchange flow

Baroclinicity is the largest contributor to the exchange inflow in the CIS (Figs. 4, 5), reflecting buoyancy input from rivers (internal) and from the open boundary (external). To isolate its impact we use the baroclinic run (B(I+E)) with tides and wind switched off, i.e.,  $Y^B \equiv Y_B$  (Eq. 3). As a driver we adopt the along–estuary salinity gradient  $(\partial_x S)^B$  (Methods), computed from B and proportional to the baroclinic pressure gradient. First, we pair the subtidal time series of  $Q_{\rm in}^B$  and  $(\partial_x S)^B$  (Fig. 6a,c,e) and estimate the section specific lag that maximizes the Pearson correlation (Fig. 6b,d,f). We do this because the exchange flow does not adjust instantaneously to changes in the baroclinic pressure gradient; the lag can depend on the forcing timescale and varies systematically along the estuary (Dijkstra, 2024). The positive lags indicate that changes in the salinity gradient (driver) precede adjustments in the exchange flow (response).

At all sections, the baroclinic inflow  $Q_{\rm in}^B$  (black, left axis) co-varies with the along-estuary salinity gradient  $(\partial_x S)^B$  (blue, right axis). The interior section (Reloncaví) shows the smallest inflows (order  $10^4$  m $^3$  s $^{-1}$ ) and an almost in-phase response (lag +0.5 d). Mid-estuary (Desertores) exhibits a clear seasonal co-variation with a delay (+12.1 d). Near the mouth (Corcovado) inflows are largest (order  $10^5$  m $^3$  s $^{-1}$ ) and the adjustment is slower relative to the interior (+19.3 d). Winter (e.g., around July-August) reductions in  $(\partial_x S)^B$  coincide with depressed  $Q_{\rm in}^B$ , most notably at Corcovado. Using these section-specific lags, the lagged scatter of  $Q_{\rm in}^B$  versus  $(\partial_x S)^B$ , colored by  $\Delta S^B$ , is positive at all locations, with Pearson correlations r=0.63 (Reloncaví), r=0.85 (Desertores), and r=0.59 (Corcovado).

With those lags, we construct driver–response curves by binning  $(\partial_x S)^B$  and plotting responses for  $Q_{\rm in}^B, \Delta S^B$ , and  $(Q_{\rm in}\Delta S)^B$ . At Reloncaví (a–c),  $Q_{\rm in}^B$  increases only weakly across the driver range, while  $\Delta S^B$  decreases slightly toward larger gradients; consequently,  $(Q_{\rm in}\Delta S)^B$  remains nearly flat. At Desertores (d–f),  $Q_{\rm in}^B$  grows monotonically with  $(\partial_x S)^B$  and  $\Delta S^B$  shows a mild increase, so the landward salt transport strengthens mainly via the inflow branch. At Corcovado (g–i), both  $Q_{\rm in}^B$  and  $\Delta S^B$  rise with  $(\partial_x S)^B$ , yielding a pronounced amplification of  $(Q_{\rm in}\Delta S)^B$  at higher gradients.

Figure 6. Left column: Time series of baroclinic inflow  $Q_{\rm in}^{\rm B}$  (black;  $10^4~{\rm m}^3~{\rm s}^{-1}$ , left axis) and the along-estuary salinity gradient  $(\partial_x\,{\rm S})^{\rm B}$  (blue;  $10^{-3}~{\rm g~kg}^{-1}~{\rm km}^{-1}$ , right axis) at (a) Reloncaví, (c) Desertores, and (e) Corcovado during 2018. Right column: Scatter plots of lagged  $Q_{\rm in}^{\rm B}$  versus  $(\partial_x\,{\rm S})^{\rm B}$ , colored by  $\Delta S^{\rm B}$  (g kg $^{-1}$ ), with Pearson correlation r and p-value for (b) Reloncaví, (d) Desertores, and (f) Corcovado. Positive lag indicates that  $(\partial_x\,{\rm S})^{\rm B}$  leads  $Q_{\rm in}^{\rm B}$ . Note: Axes differ by section to show the relation; nevertheless, the absolute exchange remains ordered (Corcovado > Desertores > Reloncaví) by roughly an order of magnitude from mouth to head.

During Winter, the along–estuary salinity gradient weakens, primarily influenced by conditions near Corcovado, which suppresses the baroclinic inflow throughout the system. This seasonal reduction in baroclinicity motivates the analyses in the subsequent sections, where we quantify the modulations due to tides alone (BT) and the full forcing including tides and wind (BTW).

Figure 7. Driver–response curves using horizontal salinity gradient  $(\partial_x S)^B$  as driver. Panels show binned medians (lines) and interquartile ranges (shading) of (a,d,g)  $Q_{\rm in}^B$  ( $10^4$  m<sup>3</sup> s<sup>-1</sup>), (b,e,h)  $\Delta S^B$  (g kg<sup>-1</sup>), and (c,f,i)  $(Q_{\rm in}\Delta S)^B$  ( $10^4$  g kg<sup>-1</sup> m<sup>3</sup> s<sup>-1</sup>) as functions of  $(\partial_x S)^B$  ( $10^{-3}$  g kg<sup>-1</sup> km<sup>-1</sup>). Rows correspond to Reloncaví (top), Desertores (middle), and Corcovado (bottom). Responses are shifted by the section specific lag (same Fig.8) that maximizes the correlation with the driver.

## 3.4 Tidal effects on exchange flow


Here we ask how the spring-neap cycle modulates the exchange flow on a baroclinic background, and then how large the incremental tidal response is beyond that baseline. To address the first point, we analyze the BT run (baroclinic+tides; wind off) and compare time series of  $Q_{\rm in}^{BT}$  with the spring-neap index. To quantify the incremental tidal effect on the TEF metrics, we







use the switch definition  $Y^{\text{tide}} \equiv Y_{BT} - Y_B$  (Eq. 2). Because the tide-baroclinicity interactions,  $Y^{\text{tide}}$  should be interpreted as the tidal response that acts on the baroclinic state, not as a signal 'pure tide only'.

Consistent with the annual budget (Fig. 5), tides contribute a smaller share to the exchange flow than the baroclinic background; nevertheless, a clear spring—neap modulation is evident in the BT run. The subtidal time series of  $Q_{\rm in}^{BT}$  (black) plotted with the spring—neap index (blue, 0= neap, 1= spring) ... show a fortnightly signal riding on a slower envelope; this envelope reflects tides acting on a seasonally varying baroclinic state, as already evident in the baroclinicity results (Fig. 6). The tidal imprint strengthens southward: in Reloncaví (Fig. 8a)  $Q_{\rm in}^{BT}$  fluctuates weakly around  $1\times 10^4$  m<sup>3</sup> s<sup>-1</sup> with modest spring—neap contrast; in Desertores (c) peaks reach  $\sim 4\times 10^4$  m<sup>3</sup> s<sup>-1</sup> during springs; and in Corcovado (e) spring episodes commonly exceed  $5\times 10^4$  m<sup>3</sup> s<sup>-1</sup>. Scatter plots of the isolated tidal contribution  $Q_{\rm in}^{\rm tide}$  versus the tidal index (colored by  $\Delta S$ ; Fig. 8b,d,f) confirm positive but modest correlations that increase southward ( $r=0.13,\ 0.29,\ 0.37;\ p\ll 0.01$ ). Phase is least coherent in Reloncaví, whereas Desertores and Corcovado are closer to in—phase. In those sections, larger fortnightly peaks in  $Q_{\rm in}^{BT}$  align with spring tides.

Driver–response curves (Fig. 9) reveal spatial contrasts. The exchange inflow,  $Q_{\rm in}^{\rm tide}$  is consistently negative across the spring–neap range at Reloncaví (panel a), indicating that tides predominantly reduce  $Q_{\rm in}$  at this location. In Desertores (panel d) and Corcovado (panel g),  $Q_{\rm in}^{\rm tide}$  is negative during neap and transitions to positive during spring tides, with a stronger spring enhancement at the mouth. In Desertores (panels e),  $\Delta S^{\rm tide}$  is negative during neap and turns positive toward spring tide. In this tidally energetic channel, spring tides enhance bottom inflow and shear, suggesting the import of saltier water, and the generation of residual stratification via straining of the along-channel salinity gradient. During neap, the weaker shear/straining cannot offset background buoyancy, so  $\Delta S^{\rm tide} < 0$ . In contrast, at Corcovado (panel h)  $\Delta S^{\rm tide}$  is largest during neaps and mostly weakens toward spring tide, pointing to spring–tide mixing that offsets straining, so stratification is best preserved during neaps. The net landward salt transport,  $(Q_{\rm in}\Delta S)^{\rm tide}$ , is near-zero to slightly positive at Reloncaví (panel c). The landward salt flux exhibits a pronounced spring-tide enhancement at Desertores and Corcovado (panel f,i) as the gain in  $Q_{\rm in}$  overcomes concurrent changes in  $\Delta S$ .

# 3.5 Wind effects on exchange flow

As the wind is the dominant driver of the variability in the exchange flow, particularly at high frequencies we can determine the influence of wind on exchange inflows using a comparison of time series from the full model run,  $Q_{in}^{BTW}$ , and  $\tau_y$  ((Fig. 10). This comparison indicates that episodic wind can influence the exchange flow at all sections. In Reloncaví,  $Q_{in}^{BTW}$  fluctuates around  $2-4\times10^4\,\mathrm{m}^3\,\mathrm{s}^{-1}$  and is enhanced with southward or down-estuary winds (winter-spring) and attenuated with northward or up-estuary winds (summer). In Desertores  $Q_{in}^{BTW}$  reaches peaks of  $\sim 8\times10^4\,\mathrm{m}^3\,\mathrm{s}^{-1}$  with a similar relationship to  $\tau_y$  as Reloncaví. Finally, Corcovado attains the largest exchange inflows ( $Q_{in}^{BTW} > 1.5\times10^5\,\mathrm{m}^3\,\mathrm{s}^{-1}$ ) during intense fall-winter southward or down-estuary wind events (Fig. 10a,c,e). Scatter plots of  $Q_{in}^{BTW}$  versus  $\tau_y$  (colored by  $\Delta S$ ; Fig. 10b,d,f) yield negative correlations that strengthen southward ( $r\approx-0.44, -0.50, -0.77$  for Reloncaví, Desertores, Corcovado;  $p\leq0.01$ ), consistent with enhancement and suppression of the exchange inflow with down-estuary and up-estuary winds, respectively.

**Figure 8.** Left column: Subtidal time series of  $Q_{\rm in}^{\rm BT}$  (m<sup>3</sup> s<sup>-1</sup>, blue) and spring-neap tidal index (dimensionless, black) at (a) Reloncaví, (c) Desertores, (e) Corcovado in 2018. Right column: Scatter plots of  $Q_{\rm in}^{\rm BT}$  vs tidal index, colored by  $\Delta S$  (g kg<sup>-1</sup>), with correlation r and p-value for (b) Reloncaví, (d) Desertores, (f) Corcovado.

Salinity differences,  $\Delta S$  are also enhanced when the exchange inflow is enhanced with down-estuary (southward) wind events, while  $\Delta S$  reduces, and even indicates that  $S_{out}$  is smaller than  $S_{in}$  with up-estuary (northward) winds.

We isolate the impact of the wind on the exchange flow as the difference between the full model run and the baroclinic + tide run, i.e.,  $(\cdot)^{\text{wind}} \equiv (\cdot)^{\text{BTW}} - (\cdot)^{\text{BT}}$  (equation 1), with along-channel wind stress  $\tau_y$  defined positive northward (up-estuary). Driver response curves depicting the influence of wind stress  $(\tau_y)$  on the isolated wind-driven exchange inflow  $(Q_{in}^{wind}, \text{ equation } XX)$  show four main features (Fig. 11). First, Down-estuary (southward,  $\tau_y 

Figure 9. Driver-response curves for tides effects, showing binned medians (lines) with interquartile ranges (shading) on tidal index (neap 0 to spring 1) for (a,d,g)  $Q_{\rm in}^{\rm tide}$  (m<sup>3</sup> s<sup>-1</sup>), (b,e,h)  $\Delta S^{\rm tide}$  (g kg<sup>-1</sup>), (c,f,i)  $(Q_{\rm in}\Delta S)^{\rm tide}$  (g kg<sup>-1</sup> m<sup>3</sup> s<sup>-1</sup>) at Reloncaví (top), Desertores (middle), Corcovado (bottom).

 $(Q_{\rm in}\Delta S)^{\rm wind}$  at all sections (panels c,f,i). Only in Reloncaví does a moderate down-estuary wind  $(\tau_y \approx -0.10\,{\rm N\,m^{-2}})$  also sharpen stratification, yielding positive  $\Delta S^{\rm wind}$  (panel b). Second, stronger down-estuary and up-estuary winds reduce  $\Delta S$ , consistent with wind stirring acting within a specific limits. (Scully et al., 2005). Third, the same down-estuary wind stress amplifies  $Q_{\rm in}^{\rm wind}$  by more than  $\sim 2.7 \times$  from Reloncaví to Corcovado (panels a vs. g), whereas the gain in  $(Q_{\rm in}\Delta S)^{\rm wind}$  is slightly smaller ( $\sim 2.5 \times$ , panels c vs. i), due to the decrease of  $\Delta S^{\rm wind}$  from Reloncaví to Corcovado (panels b,e,h). Finally,

**Figure 10.** Left: Subtidal time series of  $Q_{\rm in}^{\rm BTW}$  (black, m<sup>3</sup> s<sup>-1</sup>) and along-channel wind stress  $\tau_y$  (red arrows for direction) at (a) Reloncaví, (c) Desertores, (e) Corcovado in 2018. Right: Scatter plots of  $Q_{\rm in}^{\rm BTW}$  vs  $\tau_y$  (N m<sup>-2</sup>), colored by  $\Delta S$  (g kg<sup>-1</sup>), with correlation r and p-value for (b) Reloncaví, (d) Desertores, (f) Corcovado.

outside the wind straining window,  $\Delta S^{\mathrm{wind}}$  decreases for  $\tau_y < 0$  when winds intensify and decreases roughly exponentially with  $\tau_y > 0$  (up-estuary or northward wind), pointing to enhanced near-surface mixing due to wind stress as a dominant control on stratification. Overall, wind acts as an episodic modulator: down-estuary wind enhances  $Q_{\mathrm{in}}$  (most strongly at the mouth) and boosts net salt import, whereas up-estuary wind frequently damps  $Q_{\mathrm{in}}$  and  $Q_{\mathrm{in}}\Delta S$  through mixing. These asymmetries and their spatial gradients set up the discussion section of this manuscript, which will elaborate on competing mechanisms (straining vs. stirring) in controlling the exchange flow and salt flux in the CIS.

Figure 11. Driver-response curves for wind effects, showing binned medians (lines) with interquartile ranges (shading) on (a,d,g)  $Q_{\rm in}^{\rm wind}$  (m<sup>3</sup> s<sup>-1</sup>), (b,e,h)  $\Delta S^{\rm wind}$  (g kg<sup>-1</sup>), (c,f,i)  $(Q_{\rm in}\Delta S)^{\rm wind}$  (g kg<sup>-1</sup> m<sup>3</sup> s<sup>-1</sup>) vs  $\tau_y$  (N m<sup>-2</sup>) at Reloncaví (top), Desertores (middle), Corcovado (bottom). Arrows indicate down-wind (blue) and up-wind (red) tendencies.

## 4 Discussion

In a multi-basin Patagonian fjord system, we have quantified the relative contribution of rivers, tides, winds, and remote density variability to total exchange flows (TEF) and landward salt flux ( $Q_{\rm in}\Delta S$ ). The salt flux underpins the exchange of tracers between the fjords and ocean, and therefore the ability of the interior deep waters of the fjords and channels to be renewed






(Liungman et al., 2001). Results showed that internal baroclinicity (rivers) establishes the baseline exchange, but modulating factors (tides and winds) reshape the circulation patterns through which salt enters the fjord basins. Below, we delve deeper into the mechanics behind each driver considered and its impact on exchange flow and determine the dominant timescales over which TEF varies. Finally, we place our findings in the context of projected 21st-century changes and discuss the limitations of our study.

#### 4.1 Mechanistic controls on exchange flow

Our numerical experiments show that internal baroclinicity, that is primarily driven by river discharge, sets the baseline for the exchange flow in the Chiloé Inner Sea (CIS). It contributes roughly 45–55% to the root mean square (RMS) variability of  $Q_{\rm in}$ ,  $\Delta S$ , and  $Q_{\rm in}\Delta S$  across all sections (Fig. 5). This dominance is consistent with classical estuarine theory, in which buoyancy gradients sustain a two-layer exchange (Hansen and Rattray, 1965). A key result of our study is that winds and tides act as modulators of this baseline in two distinct ways. They either act as agents of stirring that mix and weaken stratification, or, under suitable background conditions, act to enhance stratification via wind or tidal straining. Our results indicate that, for certain ranges of tidal energy and wind stress, these straining pathways can amplify the buoyancy-driven flow, thereby modulating the baseline exchange flow set by rivers.

By contrast, external baroclinicity arises from density anomalies imposed at the ocean boundary that induce along–channel pressure gradient adjustments. In our analysis this forcing is secondary in terms of variance (10–25% RMS at Corcovado), yet dynamically relevant in winter, when fresher waters offshore (Guafo/Corcovado) decrease, or even invert, the along channel density gradient (Fig. 2e), thereby weakening the buoyancy-driven exchange and occasionally reversing it. Circulation driven by external baroclinicity has been found to be a first–order driver of fjord–shelf exchange flows in Greenlandic and Scandinavian deep-silled fjords (Jackson et al., 2018; Sutherland et al., 2014; Arneborg et al., 2004; Inall et al., 2015). In contrast, its role remains underexplored in Chilean Patagonia. Our results indicate that boundary driven baroclinic adjustments can coexist with river or wind-driven flows. However, fully resolving their importance, intermittency, and coupling to other TEF drivers requires specific observations and model experiments that are outside of the scope of this study.

Relative to river and wind forcing, tides account for a smaller share of subtidal variability in our domain (Fig. 5; 10–20% RMS ), yet their imprint is spatially structured and conditioned by the background baroclinicity. Over the fortnightly cycle, stronger spring tides generally enhance the exchange inflow  $(Q_{\rm in})$  without a dramatic collapse of  $(\Delta S)$  and as a result their is landward salt import  $Q_{\rm in}\Delta S$  increases (Fig. 9). Using the same TEF metrics, MacCready and Geyer (2024) showed in the Salish Sea that stronger tides typically raise  $Q_{\rm in}$  but reduce  $\Delta S$  enough that  $Q_{\rm in}\Delta S$  often declines, and interpreted the spring enhancement of  $Q_{\rm in}$  as being controlled by tidal pumping. By analogy, the spring-tide increase of  $Q_{\rm in}$  we find, especially in Corcovado and Desertores, could reflect a similar mechanism; however, the lack of a minimization of  $\Delta S$  suggests that shear-straining processes (tidal straining) may also contribute. Tidal straining is the ebb-flood asymmetry by which tidal currents strain the horizontal salinity (density) gradient, such that ebb tides strengthen stratification while flood tides weaken it. This can create a residual flow with enhanced bottom inflow and a compensating surface outflow(Simpson et al., 1990; Stacey et al., 2008; Burchard and Hetland, 2010), like what we see in the CIS.






Wind provides the second–largest contribution to exchange flow variability in CIS (Fig. 5, 20-50%), with a clearer imprint in fall-winter. Dynamically, wind acts through two pathways: a barotropic sea–level slope (set–up/set–down) that drives depth–uniform flow, and wind straining, where along–channel surface shear stress strains the horizontal salinity (density) gradient. In the latter case, down–estuary winds ( $\tau_y < 0$ ) promote stratification, whereas up–estuary winds ( $\tau_y > 0$ ) erode it via enhanced vertical mixing (Scully et al., 2005). In Reloncaví, the exchange flow behaves as a coupled buoyancy–wind system where rivers set a high baseline  $\Delta S$  that closely tracks discharge (Figure 6a), while episodic down–estuary winds raise exchange inflow  $Q_{\rm in}$  (Figure 11a). Within a specific range of moderate down-wind stress ( $\sim$ 0–0.10 N m<sup>-2</sup>), we observe a response consistent with wind straining ((Scully et al., 2005)), where  $\Delta S$  increases before turbulence at higher stresses reduces it, which is a behavior expected for partially mixed estuaries (Chen et al., 2012). The net result is a compensating regime sustaining landward salt import  $Q_{\rm in}\Delta S$  across seasons. In particular, when  $Q_{\rm in}$  weakens, high  $\Delta S$  (set by buoyancy) maintains import, and when  $\Delta S$  relaxes, down–estuary winds boost  $Q_{\rm in}$ . When winds blow out–estuary, the wind-strained circulation reinforces the classical buoyancy-driven pattern by enhancing the surface outflow of freshwater, which can be partially masked by the baseline signal.

The local wind behavior sits within a regional context where the wind's imprint on exchange is ubiquitous. The Southern Hemisphere westerlies and the seasonal southward migration of the Southeast Pacific Subtropical Anticyclone set a strong, seasonally modulated wind regime and synoptic variability along 40–56°S (Pérez-Santos et al., 2019). Valle-Levinson and Blanco (2004) at sill-constrictions in Patagonian fjords, found that wind events drive barotropic setup/setdown and depth-uniform reversals on 2–7 day timescales that substantially modify the mean two-layer exchange by opposing estuarine exchange and reducing flushing during events. (Valle-Levinson and Blanco, 2004; Salinas et al., 2007). In Reloncaví, observations show wind-driven reversals during up-fjord winds and low discharge, consistent with numerical results where strong events erode  $\Delta S$  (Castillo et al., 2012; Pinilla et al., 2024). These wind events also excite low-frequency internal seiches that modulate shear and intermittently mask the buoyancy signal (Castillo et al., 2017). Further south, analyses in Moraleda Channel show along-channel wind stress organizing the vertical structure of the residual circulation including transitions to three-layer flow during wind episodes (Soto-Riquelme et al., 2023). Taken together, Patagonia exhibits wind–dominated modulation of exchange through barotropic set–up, reversals, and internal seiching. Here we additionally document a wind–straining, consistent with an increase in stratification under moderate down–estuary winds in Reloncaví, clarifying the compensation between  $Q_{\rm in}$  and  $\Delta S$ .

#### 4.2 Timescales of oscillation

A wavelet analysis was used to determine the energy bands that are most prominent in the subtidal variability of the exchange inflow,  $Q_{\rm in}$ , and to glean insights into the larger climatic phenomena that can cause such variability (Fig. 12). Over the year of 2018 the time series of the exchange inflow driven by tides, winds, and baroclinicity each qualitatively show distinct periodicities of variability (Figure 4). The periodicities that were found to dominate the exchange inflows in all cases are the synoptic band (3-12 d), the fortnightly band (14-16 d), the monthly band (25-35 d) and a lower-frequency band (40-70 d) (not shown).

We will now discuss the forcing mechanisms of each of these bands and how they manifest in the exchange inflows.


**Figure 12.** Wavelet power of subtidal  $Q_{\rm in}$  by section, band, and experiment during 2018. Panels are lettered (a–l). Rows (period bands): (a–c) 3–12 d; (d–f) 14–16 d; (g–i) 25–35 d; (j–l) 40–70 d. Columns (experiments; Methods): (a,d,g,j) BTW = full forcing; (b,e,h,k) BT = baroclinicity+tides; (c,f,i,l) B = baroclinicity-only. In each panel, colors indicate  $\log_2$  wavelet power (darker = stronger); the x-axis spans Jan–Dec 2018. Sections are stacked vertically (Reloncaví, Desertores, Corcovado); dashed lines separate sections.

Energy in the 3–12 d band is primarily wind–driven (storm passages) (Fig. 12a-c). The pattern is consistent across sections with the wind–inclusive case  $Q_{\rm in}^{\rm BTW}$  (panel a) showing the largest power, and weakening in when only tides+baroclinicity are retained  $Q_{\rm in}^{\rm BT}$  (panel b), and smallest for baroclinicity alone  $Q_{\rm in}^{\rm B}$  (panel c). This ordering attributes the 3–12 d variability primarily to storms and agrees with prior research in Patagonia that has found a period of 2–7 d for wind set–up, depth-uniform reversals, and internal seiching modulating exchange in Reloncaví and Moraleda (Valle-Levinson and Blanco, 2004; Castillo et al., 2012) and synoptic wind control on residual circulation in nearby channels (Soto-Riquelme et al., 2023; Pinilla et al., 2024, 2020a).





Fortnightly scales of variability intensify toward the ocean (Corcovado and Desertores), consistent with stronger tidal currents (Fig. 12d-f and Fig. 3c). Interestingly, at Corcovado, there is elevated power at the 14-16 d periodicity for the exchange inflow with baroclinicity alone ( $Q_{\rm in}^{\rm B}$ ). This implies that there are some nonlinear interactions between tidal currents and baroclinicity over the spring-neap cycle at Corcovado that influence the exchange inflow.

Energy in the 25–35 d band exhibits a broader distribution across the different forcings than the previously analyzed frequencies. Specifically, when forcings such as wind and tides are removed, energy remains more conserved at this frequency band than at higher frequencies. We associate this band with the Southern Hemisphere *Baroclinic Annular Mode* (BAM), which is a tropospheric pattern that organizes storminess and synoptic eddy activity on  $\sim$ 20–30 d cycles and manifests in surface winds and pressure (Thompson and Woodworth, 2014). The BAM periodicities have been found in wind velocities near the Guafo Mouth, the main inlet connecting the fjords and channels of Northern Patagonia to the Pacific Ocean, in a study by (Ross et al., 2025). Further, Ross et al., (2014) identified that winds induced by the BAM modulated the depth of the pycnocline in a fjord in Southern Chile. They investigated a time series of 10 years of atmospheric pressure at the study site that also confirmed that storminess occurs in this area at the BAM frequency. An additional study by Pérez-Santos et al. (2019) found upwelling conditions at the BAM periodicity, indicating that the BAM can influence exchange flows either through direct wind forcing or through baroclinicity, which is likely why we see elevated wavelet power in the exchange inflow driven solely by baroclinicity,  $Q_{in}^B$ , at this frequency (Figure 12i).

Energy in the low-frequency intraseasonal band (40–70 d) is elevated under all forcing configurations and remains spatially coherent across simulations (Fig. 12j-l). It is stronger in the baroclinicity–only run ( $Q_{\rm in}^B$ ) and the total run ( $Q_{\rm in}^{BTW}$ ) than in the baroclinity-tide run ( $Q_{\rm in}^{BT}$ ), and increases toward Corcovado in each case. The intraseasonal timescales of 40-70 d are consistent with the Madden–Julian Oscillation (MJO) teleconnections previously detected in Patagonia (Ross et al., 2025; Pérez-Santos et al., 2019; Jacques-Coper et al., 2023, 2015). Ross et al. (2025) reported a dominant 40–70 d periodicity modulating subtidal along–channel velocity and the barotropic pressure gradient in the Guafo Mouth, hypothesizing an MJO pathway in which equatorial Kelvin waves become coastal trapped waves along the South American margin, taking  $\sim$ 43–56 d to transit from the equator to Chilean Patagonia. Consistently, MJO activity modulates extremes and biological responses in the region, including phytoplankton variability in the Inner Sea of Chiloé (Jacques-Coper et al., 2023).

Importantly, our open boundary condition setup prescribes only a seasonal T–S cycle (no explicit intraseasonal coastal or remote forcing). Thus, the observed 40–70 d variance must arise from local processes such as low-frequency envelopes of along-channel wind or river discharge, rather ocean forcing. This contrasts with Ross et al. (2025) as our design deliberately cannot test that remote pathway, but future studies should consider it. The following section will elaborate more on how the timescales of variability found to be important in modulating the exchange flow in the CIS will respond to anticipated climate change.

#### 4.3 Future climate context and implications for exchange flow

Consistent large-scale signals indicate regional warming and, seasonally, a tendency toward a stronger and more persistent subtropical anticyclone over the southeast Pacific (Salazar et al., 2024). In northern Patagonia (approximately 40–45°S), multi-





model CMIP6 assessments suggest likely decreases in mean precipitation (scenario-dependent and stronger under SSP5–8.5) (Salazar et al., 2024). In the CIS watershed, this combination favors reduced mean precipitation and runoff and longer low-flow periods (Aguayo et al., 2021), while surface pressure patterns more often support weak or up-estuary winds; observed and modeled links between a strengthened subtropical high and regional circulation lend dynamical plausibility to this tendency (Garreaud et al., 2021; Boisier et al., 2016). Modes of variability, including the Southern Annular Mode (SAM) and El Niño–Southern Oscillation (ENSO), can modulate these trends, with positive SAM phases promoting drier conditions in northern Patagonia and up-estuary winds, and reducing seasonal runoff (León-Muñoz et al., 2018).

In our findings, both shifts act in the same direction: a lower baseline  $Q_R$  weakens the salinity contrast  $\Delta S$  sustained by buoyancy input, and weak or up–estuary winds further erode  $\Delta S$ ; in practice these often co-occur (persistent subtropical-high conditions bring reduced precipitation/runoff and favor up–estuary winds (Garreaud et al., 2013)), so salt import  $Q_{\rm in}\Delta S$  diminishes even when  $Q_{\rm in}$  remains sizable. This anticyclonic, dry background therefore links directly to lower  $Q_{\rm in}\Delta S$  and a reduced likelihood of deep-water renewal. By contrast, episodes with moderate down–estuary winds frequently coincide with elevated runoff, which helps preserve  $\Delta S$ ; under this co-occurrence the two components of salt import align, yielding higher  $Q_{\rm in}\Delta S$  and the most favorable windows for renewal.

#### 4.4 Limitations

The way we isolate wind, tides, and baroclinicity in our numerical experiments yields a total influence for each driver that necessarily includes the product of its interactions with the other forcings. This is useful for mechanism discovery because it reveals emergent wind–tide–buoyancy behavior, but it does not quantify a "pure" main effect; rather, it shows how one driver modulates the coupled system. Prior work demonstrates that wind–tide–buoyancy coupling can produce residuals that violate linear superposition (Guo and Valle-Levinson, 2008; Burchard and Hetland, 2010; Burchard et al., 2011). Consequently, the "annual percent contributions" in Fig. 5 should be interpreted as modulation capacity under co-acting forcings, not as additive shares of the full signal.

We prescribe seasonal temperature and salinity at the ocean boundary and harmonic tides, while this facilitates isolate the internal processes, it underestimates the role of remote forcings or wave propagation that could alter stratification and flow patterns near the boundaries. This includes omitting intraseasonal barotropic set-up, like a costal trapped waves and upwelling/downwelling ocean conditions (Ross et al., 2025; Poli et al., 2022; Strub et al., 2019). However, it is noteworthy that our results show strong intraseasonal signals at 25–30 days and 40–70 days, especially at the southern border of Corcovado, indicating that these fluctuations originate from large-scale modes but have local impacts specifically through wind and rivers. Therefore, the ensemble of remote versus local processes and scales is a task that warrants attention in future studies.

## 5 Conclusions

This study aimed to determine how rivers, wind, tides, and boundary baroclinicity shape exchange flow along the multi-basin Chiloé Inner Sea (CIS). To reach this goal we analyzed two complementary TEF metrics: the exchange inflow  $Q_{\rm in}$  and the salt






import  $Q_{\rm in}\Delta S$ , where  $\Delta S$  is the TEF dynamic stratification between inflow and outflow classes. Influence of river runoff (also coined internal baroclinicity) explained 45-55% of the variance of  $Q_{\rm in}$ ,  $\Delta S$ , and  $Q_{\rm in}\Delta S$  along the CIS. Winds and tides were found to modulate the baseline exchange flow established by internal baroclinicity. Down-estuary winds enhanced  $Q_{\rm in}$  and salt import  $(Q_{\rm in}\Delta S)$  system-wide when moderate down-estuary wind stress coincided with high river discharge, preserving stratification, especially at the most upstream considered section. Persistent up-estuary winds or strong wind events of either sign were found to erode  $\Delta S$ , causing  $Q_{\rm in}\Delta S$  to subsequently drop even when  $Q_{\rm in}$  remained large. The import of dense salty water is critical to deep-water renewal in fjords, and these results indicate that renewal events can be hindered by up-estuary directed winds or strong wind events in general.

Beyond synoptic wind–river control, seasonal density anomalies originating from the ocean boundary intensify down-fjord (at Corcovado) in winter. At times the external baroclinicity can modulate the internally driven exchange and attenuate, or even reverse, the local exchange flow regime. This underscores the need to monitor boundary temperature and salinity alongside wind and rivers to fully capture exchange flow dynamics in fjord systems. The observed organization of TEF into synoptic (3–12 d), fortnightly (14–16 d), monthly (20-30 d), and intraseasonal (40–70 d) bands, documented here for one year, indicate that forecasts and observing systems should resolve these timescales and be evaluated over multiple years to fully capture the exchange flow and salt flux between the ocean and inner fjords and channels. Because the TEF metrics used here are standard and comparable across estuaries, these insights are applicable to geometrically similar fjords when local wind, runoff, tidal range, and boundary stratification fall within analogous regimes.

Data availability. The raw data supporting the conclusions of this article will be made available by the authors

Author contributions. EP: Conceptualization, Data curation, Formal analysis, Investigation, Methodology, Software, Validation, Visualization, Writing – original draft, Writing – review and editing. LR: Conceptualization, Formal analysis, Funding acquisition, Investigation, Methodology, Project administration, Resources, Supervision, Validation, Writing – original draft, Writing – review and editing

485 *Competing interests.* The authors declare that the research was conducted in the absence of any commercial or financial relationships that could be construed as a potential conflict of interest

Acknowledgements. This research was supported by the National Science Foundation under Grant No. 2045866. The authors extend their gratitude to the 'Chonos' Initiative of the Instituto de Fomento Pesquero (IFOP), Chile, for providing access to its computational resources, enabling the use of the MIKE 3, WRF, and FLOW-IFOP models for this study. We also thank the model developers Oliver Venegas for his contributions to the WRF hindcast and Mark Falvey for the FLOW-IFOP hydrological model. Additionally, we thank Iván Pérez-Santos for

sharing information from the OMARE buoy for models validations. Lauren Ross also received a Fulbright U.S. Scholar grant during the research that led to this publication.

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
