# Peer review of "Disentangling Physical Forcings Influencing Exchange Flow in a Multi-basin Fjord System: Chiloé Inner Sea, Patagonia"

_EGUsphere, 2025_

## Referee Comment (RC1)

Review of "Disentangling Physical Forcings..." by Pinilla and Ross.

This modeling study explores the effects of various forcing mechanisms (tides, winds, river runoff, etc.) on the two-layer exchange in the Chloé Inner Sea, a large and complex estuary system in Patagonia. The primary metrics are the volume rate of inflow of salty ocean water, the corresponding salt flux, and the broad salinity gradient, all measured at different cross sections. The main forcing mechanisms affecting different locations, or acting over different time scales, are identified, all with the caveat that nonlinear interactions can make it difficult to assign a specific forcing to a specific response. I am not an expert on fjord or estuary circulations, but it appears that the work will be an important contribution to the understanding of the physics of the Chiloé Inner Sea and its response to climate change.

I found the paper to generally be easy to read, with good graphics and explanations, though I have identified a number of spots that need some clarification. My recommendation is for moderate revision.

**General points:**

Are there are novel processes that make this application unique or different within estuaries and marginal seas in general? If so, these are things the authors might wish to emphasize in their final version.

For example, marginal seas that are strongly forced can be driven into a state of maximal exchange, an idea first floated by Stommel and Farmer (1952). They used a two-layer estuary system as an example, but it has turned out that the main applications are the Mediterranean Sea and the ancient Red Sea. Maximal exchange occurs when the inflow/outflow is choked by some narrow passage (the Desertores section, perhaps?), and the mixing within the Ancud Gulf is strong enough to drive the flow to it maximum exchange limit, determined by hydraulics. More generally, is there hydraulic activity in the Desertores Pass?

One other general topic that I want to ask the authors about: Mixing seems crucial to metrics such as $Q_{in=0}$ and $\Delta S_{in}$. For example, a run performed with only river runoff and with no mixing allowed would end up with $Q_{in=0}$. Wind and tides contribute to mixing, and when these are shut off, the mixing generally decrease. So the results shown in Figure 4, which suggest that the magnitude of $Q_{in=0}$ and $\Delta S_{in}$ are largely captured in experiments where wind and tides are shut off, suggests: 1) that mixing is coming from a source other than winds and tides, or 2): that even though winds and tides are absence, the turbulence parameterization in the model are somehow retaining tides and wind as energy sources, even though tides and wind have been turned off. I know that scenario #2 has been a

focus of concern in other types of modeling studies, usually where the wind has been turned on and off, and that some modelers will decrease their eddy coefficients when the wind is turned off. What is the situation here?

Specific points:

Sec. 2.3.2 The partitioning into $Q_{in}$ and $Q_{out}$ seems to presuppose a 2-layer system. Sometimes people will define an "interfacial" layer, one that is created by mixing between upper and lower layers. Does the actual stratification in the model generally look like at 2-layer system?

Eqs. (1) – (4). The readers question on seeing these is why is there not $Y^B$-$Y^{B(E)}$?

Line 93: states that tidal current of up to 4m/s occur in Reloncav Sound (should it be Reloncavi?) but Fig. 3c suggests otherwise.

Line 190: Has stratification "index" been defined?

Fig. 6. It appears to me that the lagged correlation between the salinity gradient and inflow would be much higher if the first 3 months or so of the record were excluded. In Jan-March, the two time series seem to be anticorrelated. What is happening during this period?

Line 273. Are you referring to Fig. 8 here?

Figure 9. As mentioned in the text, the left-hand panels suggest that $Q_{in}$ is enhanced at the Desertores and Corcovado sections during periods of spring tides, which is consistent with greater mixing. This makes sense. During neap tides, $Q_{in}$ is decreased at all three sections, and the idea put forth here is that the tides are too weak to cause any mixing and that they are inhibiting the inflow through some other mechanism. The accompanying text (lines 280-290) don't make it clear what that mechanism is. Can something be said?

Figure 10. The wind arrow in the upper left panel is labeled upwind/downwind. Since the wine changed direction seasonally, does this mean that a positive wind stress is directed southward in fall and winter, and northward in spring and summer? Or should the label on the wind vector be "down estuary/up estuary"? Same comment for Fig. 11.

Line 307. Eq. XX ?

Line 313. There has been some mention of wind straining, and here the "wind straining window". I'm not sure I understand what the authors are referring to: does "straining" refer to the horizontal divergence of the flow due to the combination of wind and boundaries? This issue comes up again on line 334, where tidal and wind straining is said to increase stratification. As I see it, straining in the horizontal can certainly cause changes in the

horizontal salinity gradient and this, in turn, could affect salinity gradients in the vertical.  I assume this is what you are talking about, but I want to make sure it is clear to the readers.

(When we get to lines 353-354 we seem to have a definition of tidal straining: here it is said that flood tides tend to weaken stratification and ebb tides tend to weaken it. Hmmm. Perhaps this should be explained further, at least for those of us who are not estuary experts.)

Lines 310-311.  I'm not sure where the 2.7 comes from. The maxim amplification at Reloncavi is about 2.0,  whereas the maximum at Corcovado is 9.5.  A comparison of the two would give an amplification of 9.5/2, so the authors must be referring to something else.

Line 316.  It is stated that up-estuary winds frequently damp $Q_{in}$ through mixing, but an increase in mixing alone should tend to increase the volume exchange rate, and thus $Q_{in}$. I would have thought that the reason a strong, up-estuary wind would tend to damp $Q_{in}$ is that it is essentially dragging the upper layer up into the estuary.  So this really seems like momentum/pressure gradient effect.

Line 399-401.  Elevated power at 14-16 days is found at Corcovado for a model run that does not include tides. This is attributed to nonlinear interactions between tidal currents and baroclinicity over the spring-neap cycle.  I don't understand how a model run without tides can capture nonlinear interactions with tides.